# Tucker Decomposition for interpretable Neural ordinary differential equations

**Dimitrios Halatsis,** **Joao Pereira, Michael Alummoottil**
Imperial College London
`{d.halatsis, joao.pereira17, michael.alummoottil21}@imperial.ac.uk`

**Grigorios Chrysos**
University of Wisconsin-Madison
`chrysos@wisc.edu`

## Abstract

Tensorial and polynomial networks have emerged as effective tools in various fields, particularly for modeling the multilinear relationships among input variables. More recently, polynomial networks factorized using the canonical polyadic tensor decomposition (CPD) have been successfully used in the problem of system dynamics identification, where the relations between variables are usually a polynomial function. This paper introduces a more general tensorial network that employs Tucker decomposition, thereby providing enhanced flexibility and expressivity in model construction. The study evaluates the performance of TuckerNet, comparing it against CPD-based networks in learning functions and identifying ordinary differential equation dynamics. The findings demonstrate the potential of TuckerNet as a superior alternative for tensorial network construction, particularly when constraining the number of parameters, while also highlighting aspects beyond decomposition that impact learning outcomes.

## 1 Introduction

In the area of system identification traditionally mechanistic models were used to describe the dynamics. Those models were built from physical principles and iteratively improved by testing them with physical experimentation. Then tools like SINDy (Brunton et al., 2016) were introduced where the dynamics could be learnt from data by solving a sparse regression problem. More recent works (Fronk & Petzold, 2023) tackled the problem from the area of Neural ODEs (Chen et al., 2018) by exploiting the strengths of tensorial and polynomial networks (Chrysos et al., 2022; Panagakis et al., 2021).

Tensorial and polynomial networks (TNs and PNs) are a particularly valuable tool for learning ODEs, offering the ability to precisely capture system dynamics and provide symbolic equations, thereby enhancing model interpretability. This efficacy is attributed to their capacity to learn intricate functions while adhering to the simplicity and elegance of polynomial outputs, without the need for activation functions, batch normalization layers (Ioffe & Szegedy, 2015), or other deep learning tools that may obfuscate model transparency. While these tools are indispensable in domains like computer vision and NLP, and therefore most recent works use TNs as augmentation blocks in existing architectures with activation functions (Hu et al., 2018; Georgopoulos et al., 2020; 2021; Chrysos et al., 2021a; Babiloni et al., 2021; Yang et al., 2022; Chrysos et al., 2023; Cheng et al., 2024), in the topic of neural ODE learning, system dynamics can often be adequately approximated by polynomials, making PNs and TNs an ideal fit for such applications.

The effectiveness of PNs can be predominantly attributed to their capacity to represent complex, high-order functions through a tensor that can be factorized by a set of low-rank tensors, thereby circumventing the need for an exponentially increasing number of parameters and computational resources. This low-rank approximation is typically (Chrysos et al., 2021b) achieved through the canonical polyadic decomposition (CPD) (Hitchcock, 1927), wherein the output is expressed as a

---

*Some information on the author https://dhalatsis.github.io/

sum of rank-one tensors. More recent works (Kossaifi et al., 2023) have used the Tucker decomposition to factorize the convolutional weights of a Fourier Neural Operator Layer (Li et al., 2020). The Tucker decomposition (Tucker, 1966; Carroll & Chang, 1970), serving as a generalization of the CPD, can extend the expressivity of the model by enabling finer control over the decomposition properties, thus extending the model into a broader family of functions.

While the strengths of Tucker decomposition are well-documented in the context of low-rank tensor approximation (Kolda & Bader, 2009) and has been used as a form of weight factorization in neural operator learning, this study delves into its applicability for system dynamics identification, particularly in a standalone tensorial network we call TuckerNet. We assess the efficacy of TuckerNet in a simplified function learning task and compare it against CPD-based networks. Our objective is to highlight Tucker-based tensorial networks as a viable, and potentially superior, alternative to CPD-based models. Lastly, we demonstrate the effectiveness of TuckerNet in neural ODE learning by subjecting it to a higher-degree chaotic system.

## 2 METHOD

### 2.1 FORMULATION

In this section we present the TuckerNet, and give details on its complexity and properties. The idea of a tensorial network is to represent higher order interaction between input terms by a higher order tensor $\mathcal{W}^{[n]}$. In the case of Tucker, this tensor is decomposed as follows:

$$\mathcal{W}^{[n]} = \mathcal{G}^{[n]} \times_1 \boldsymbol{A}_1 \times_2 \boldsymbol{A}_2 \times_3 \ldots \times_{n-1} \boldsymbol{A}_{n-1} \times_n \boldsymbol{A}_n \times_{n+1} O = \mathcal{G}^{[n]} \times_{n+1} O \prod_{j=1}^n \times_j \boldsymbol{A}_j \ . \quad (1)$$

Where the learnable parameters are $\boldsymbol{A}_k \in \mathbb{R}^{d \times h_k}$, $\boldsymbol{O} \in \mathbb{R}^{o \times h_o}$ and $\mathcal{G}^{[n]} \in \mathbb{R}^{o \prod_k = 1^n \times_k h_k}$ with $d$ is the input dimension, $o$ is the output dimension, $h_k$ are the latent dimension for each mode, and $h_o$ is the latent output dimension. The tensor $\mathcal{W}^{[n]}$ can be used to compute the higher order function as described in Chrysos et al. (2021b). This function is essentially a polynomial. In TuckerNet will instead use the factorized version of this tensor. By substitution of the $\mathcal{W}^{[n]}$ with the equation 1. The terms can rearrange as mode-m product is commutative:

$$\boldsymbol{P}(\boldsymbol{z}) = \mathcal{G}^{[n]} \times_{n+1} O \prod_{j=1}^n \times_j \boldsymbol{A}_j \boldsymbol{z} \ . \quad (2)$$

**Implementation Note.** In order for the equation to be implemented in standard ML frameworks such as Pytorch (Paszke et al., 2017), the mode-m product is using the identity $(\mathcal{G} \times_k \boldsymbol{A}_k \boldsymbol{z})_{(k)} = G_{(k)} \boldsymbol{A}_k \boldsymbol{z}$. This requires re-matricization of the core tensor $\mathcal{G}$ on each mode during calculations, resulting in a potential computational overhead of a single matrix reshaping per mode. For the detailed algorithm please refer to the appendix.

### 2.2 PROPERTIES

The $\mathcal{W}^{[n]}$ does not have to be to be computed explicitly, instead we can directly compute the output for a given input by collapsing every mode in each iteration, as in the CPD decomposition. It is worthy to note that when $\mathcal{G}^{[n]}$ is diagonal, the TuckerNet is equivalent to the CPD decomposition (Kolda & Bader, 2009).

**Remark 2.1.** *The computational (and memory) complexity is of the TuckerNet is $O(h_o \prod_{k=1}^n h_k + d \sum_{k=1}^n h_k + oh_o)$.*

While this points towards exponential increase with respect to degree, this formulation is instead designed for finer control of the representation with increased complexity along certain modes, only when necessary.

**Interpretability Note:** One significant advantage of polynomial formulations lies in their ability to directly recover symbolic expressions from their parameters, thereby providing network transparency and some degree of interpretability. This is accomplished by computing the polynomial coefficients and subsequently reconstructing the polynomial, or equivalently, reconstructing the $\mathcal{W}^{[n]}$

Table 1: 4-variable Polynomial Performance

| model | Degree 4 | | Degree 5 | | #parameters |
|---|---|---|---|---|---|
| | Train | Test | Train | Test | |
| $\pi$-net-8 | $0.0153 \pm 0.0045$ | $1.953 \pm 0.5551$ | $989.0 \pm 208.0$ | $120446 \pm 24953$ | 484\|596 |
| $\pi$-net-12 | $0.0001 \pm 0.0001$ | $0.0020 \pm 0.0048$ | $9.677 \pm 2.249$ | $3776 \pm 1288$ | 916\|1132 |
| TuckerNet-2 | $22.53 \pm 4.596$ | $517.6 \pm 113.1$ | $16504 \pm 2499$ | $91594 \pm 177946$ | 224\|328 |
| **TuckerNet-3** | $\mathbf{0.0015 \pm 0.0013}$ | $\mathbf{0.1067 \pm 0.0840}$ | $\mathbf{4.030 \pm 4.087}$ | $\mathbf{1373.7 \pm 1781.2}$ | 564\|1272 |
| MLP | 6.864 | 56960 | - | - | 31204 |

Table 2: 5-variable degree 4 polynomial for various models

| Model | Train | Test | #parameters |
|---|---|---|---|
| $\pi$-net-10 | 212.9 | 5884 | 735 |
| $\pi$-net-15 | 0.493 | 76.50 | 1400 |
| $\pi$-net-20 | 0.004 | 0.9939 | 2265 |
| **$\pi$-net-25** | $\mathbf{10^{-7}}$ | $\mathbf{10^{-6}}$ | 3300 |
| TuckerNet-3 | 61.07 | 2925 | 765 |
| **TuckerNet-4** | **0.005** | **0.12259** | 1760 |

tensor. Remarkably, Tucker-based networks can be shown to have signficantly lower complexity for equation recovery compared to $\pi$-nets in this regard.

**Remark 2.2.** *The computational complexity for recovering the polynomial coefficients of the TuckerNet is $O(d^{n+1})$. The complexity for the $\pi$-net is $O(k^2 d^n)$.*

The increased cost of CPD reconstruction becomes evident when dealing with high degrees where the rank dominates the input dimensions, while the cost of Tucker decomposition is asymptotically constant with regards to its ranks $h_k$ as in this case $h_k \leq d$.

## 3 EXPERIMENTS

### 3.1 POLYNOMIAL LEARNING

To assess the expressive power and trainability of Tucker-based polynomial networks, we initially evaluate their strength in directly learning polynomial functions. This experiment aims to empirically compare the performance of the two decomposition methods and shed light on their parameter efficiency, for achieving a specific performance.

We generated random polynomials with coefficients sampled from $\mathcal{N}(0, 1)$ of degree $n$ and $d$ variables, evaluating them at $10^d$ uniformly spaced points within the $[-4, 4]^d$ hypercube. The choice of the $[-4, 4]$ interval stems from empirical observations indicating that polynomials sampled in this manner exhibit interesting behavior near zero, such as critical and saddle points. While moving away from zero, higher-order terms tend to dominate the polynomial. Each experiment retained a single polynomial as the target, with results averaged over 10 repetitions. Mean square error was used as the performance metric.

In Tucker polynomials, we maintained a constant latent dimension $h_k = h$ for all modes $k$, with the rank indicated by the number in the model description. For $\pi$-nets, this number corresponds to the width of the $\boldsymbol{U}_{[i]}^T$ matrices as in Chrysos et al. (2021b). This width is also equivalent with the number of rank-one tensors used in the decomposition.

As observed from the results in Table 1, TuckerNets appear to perform better than a $\pi$-net with a similar number of parameters.This difference becomes more apparent as we move into more complicated functions such as a degree 5 polynomial, where the TuckerNet-3 outperforms even the $\pi$-net-12. Another observation is that the TuckerNet is very sensitive to initialization, as we move into higher degrees the error variance increases faster for TuckerNets.

In Table 2, the advantages of Tucker decomposition become even more evident as we transition to 5-variable polynomials. Here, a rank-4 TuckerNet shows better performance compared to the $\pi$-

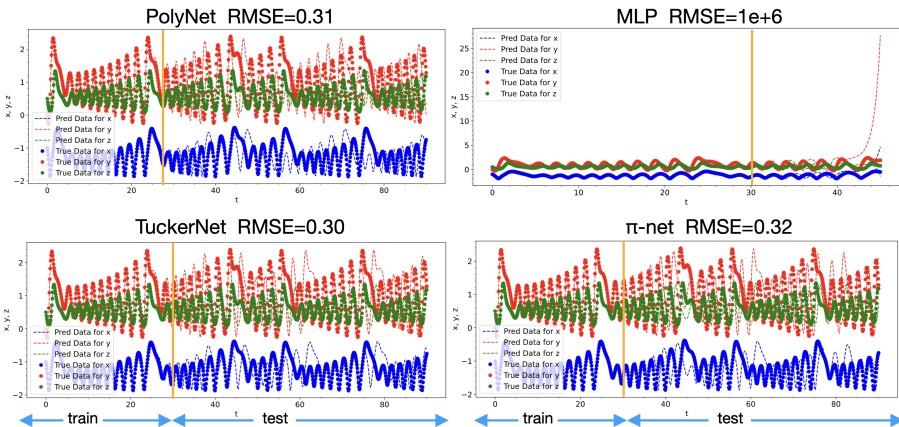

Figure 1: The different ODEs computed from training were integrated from initial conditions to time 90. The training was performed in the interval $[0, 30]$ indicated by the orange horizontal line.

net, achieving significantly better results with far fewer parameters. Additionally, we observe that beyond a certain parameter threshold, notably $\pi$-net-25, the CPD solution becomes trivial. This observation suggests that Tucker Decomposition utilizes its parameters more optimally in underparameterized models. These findings underscore the applicability of theoretical properties known in low-rank tensor approximation to machine learning settings, as confirmed by our controlled experiments.

## 3.2 ODE LEARNING

Following the evaluation on directly learning a polynomial function our next step involves employing TuckerNet to capture the behavior of dynamical systems. We selected the Rabinovich–Fabrikant equations (Rabinovich & Fabrikant, 1979) as our target ODE system. This choice was motivated by our aim to test a system of equations containing high-degree terms, ensuring sufficient complexity for discernible differences. Notably, the Rabinovich–Fabrikant equations are renowned for exhibiting chaotic behavior under certain parameter configurations.

We chose initial conditions $x = 1$, $y = -0$ and $z = 0.5$ and parameters $\gamma = 0.87$ and $\alpha = 1.1$. We generated 600 uniformly spaced data points in $t \in [0, 30]$ for integrating the IVP with the torchdiffeq Chen et al. (2018) solver using DORPI5. When then trained our models by performing a single step between two adjacent known points using fourth order Rungke-Kutta method (Fehlberg, 1970) as in Fronk & Petzold (2023). We use a TuckerNet and $\pi$-net of rank 2, a multilayer perceptron (MLP) with a hidden layer of 100 neurons and a full polynomial with all $O(d^n)$ polynomial coefficients we call PolyNet for reference.

In the results depicted in Figure 1, we observe the performance of four approaches. Notably, all three polynomial-based methods exhibit decent performance, with TuckerNet slightly outperforming the $\pi$-net, while the MLP diverges shortly after the end of training set. These plots also highlight that the challenge in learning ODEs is not solely constrained by the accuracy of the decomposition method. Even in the case of PolyNet, where no decomposition is employed, performance remains comparable. Instead we speculate that the learning method and the function sampling strategy could play crucial roles in reducing the final error.

## 4 CONCLUSIONS AND FUTURE WORK

This study introduced a tensorial formulation and showcased its effectiveness in learning neural ordinary differential equations (ODEs). While there are instances where the CPD decomposition may achieve superior accuracy, clear evidence suggests that Tucker-based approaches demonstrate an improved performance in few-parameters scenarios of the problem, where a true low-rank approximation is used. Furthermore, the experiments provided insights indicating that other factors beyond decomposition performance can bottleneck the performance of polynomial neural ODEs.

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

## A APPENDIX

### PROOFS AND ALGORITHMS

#### A.1 TUCKER NET ALGORITHM

---

**Algorithm 1** Batched Forward Pass Function for TuckerNet for Pytorch

---

**Require:** $self.G$: core tensor representing $\boldsymbol{\mathcal{G}}^{[n]}$
**Require:** $self.A$: Linear layers representing $\boldsymbol{A}_j$
**Require:** $z$: Input point $\boldsymbol{z}$
1: **function** FORWARD($t, z$)
2:     $z \leftarrow z[:, \text{None}, :]$                                  ▷ Expand dimensions of $z$
3:     $G \leftarrow \text{clone}(self.G)$                                  ▷ Clone $G$
4:     **for** $i \leftarrow 0$ **to** self.degree $- 1$ **do**
5:         $G \leftarrow G.\text{reshape}(G.\text{shape}[0], self.\text{out\_dim}, -1, self.\text{latent}).\text{permute}(0, 1, 3, 2)$
6:         $v \leftarrow self.A[i](z)[:, :, \text{None}, :]$                 ▷ Adjust dimensions of $v$
7:         $G \leftarrow vG$                                                ▷ Matrix multiplication
8:     **end for**
9:     $rv \leftarrow G.\text{reshape}(G.\text{shape}[0], self.\text{out\_dim})$        ▷ Final reshape
10:    return $rv$
11: **end function**

---

#### A.2 REMARK 2.2

#### A.2.1 TUCKER

We claim that the computational complexity for retrieving the equation for the Tucker-net is $O(d^{(n)})$, where $d$ is input dimension, $n$ the degree and $h_k$ the latent ranks. We also naturally assume that the latent ranks $h_k \leq d$. Here we will also assume that the output dimension is 1, the result can be easily extended for more.

For the Tucker decomposition scenario we need to recover the $\boldsymbol{\mathcal{W}}$ tensor. Once the $\boldsymbol{\mathcal{W}}$ is known then the formula from Chrysos et al. (2021b) can be used with symbolic variables to recover the tensor. Given the formula for computing the output of the Tucker decomposition

$$\boldsymbol{\mathcal{W}}^{[n]} = \boldsymbol{\mathcal{G}}^{[n]} \prod_{j=1}^{n} \times_j \boldsymbol{A}_j \ ,$$

Let $\boldsymbol{\mathcal{W}}_k^{[n]} = \boldsymbol{\mathcal{G}}^{[n]} \prod_{j=1}^{k} \times_j \boldsymbol{A}_j \boldsymbol{z}$, where only the first k mode-m products where applied. Then $\boldsymbol{\mathcal{W}}_k^{[n]} \in \mathbb{R}^{\prod_{j=1}^{k} \times_j d \prod_{j=k}^{n} h_j}$, and we can then describe the reconstruction process using the following recursive equation:

$$\boldsymbol{\mathcal{W}}_{k+1}^{[n]} = \boldsymbol{\mathcal{W}}_k^{[n]} \times_k \boldsymbol{A}_k \ , \tag{3}$$

Where at iteration $k$, we perform this mode-m product. This product has equivalent cost as the matricized matrix multiplication Kolda & Bader (2009) and will incur cost of $d^k h_k \prod_{j=k+1}^{n} h_j$. So the total cost for all iterations will be

$$\sum_{k=1}^{n} d^k h_k \prod_{j=k+1}^{n} h_j \leq \sum_{k=1}^{n} d^k h_{max}^{n-k} = h_{max}^n \sum_{k=1}^{n} \left( \frac{d}{h_{max}} \right)^k = O(d^{n+1}) \ .$$

#### A.2.2 $\pi$-NETV1

Here we use a different approach, where we will assume that we use symbolic variables as inputs for the $\pi$-netV1, similarily as a package like sympy would run the execution. The $\pi$-netV1 output can be defined by the following recursive relation

$$\boldsymbol{x}^{[n+1]} = \boldsymbol{C}^{[n]}((\boldsymbol{U}^{[n]}\boldsymbol{z}) * (\boldsymbol{x}^{[n]} + 1))$$

Where $*$ refers to the element-wise multiplication. $\boldsymbol{x}^{[1]} = (\boldsymbol{U}^{[1]}\boldsymbol{z})$, $P(\boldsymbol{z}) = \boldsymbol{O}\boldsymbol{x}^{[n]}$, where $U \in \mathbb{R}^{k \times d}$, $\boldsymbol{C}^{[i]} \in \mathbb{R}^{d \times d}$, $O \in \mathbb{R}^{d \times o}$ and $\boldsymbol{x}^{[i]} \in \mathbb{R}^k$, where $k$ is the rank, $d$ the input dimension and $o$ the output.

Now let assume that $z$ is a symbolic variable. The following operations will have the update effects:

- $(\boldsymbol{U}^{[n]}\boldsymbol{z})$ results in an size $k$ column vector where each element has a sum of $d$ symbolic terms. This requires $O(dk)$ operations.

- The element-wise multiplication creates new free symbolic terms. specifically for $a * b$ with $n_a$ and $n_b$ symbolic terms, this will cause approximately $n_a n_b$ symbolic terms. In our case this result will be all terms of a full polynomial of degree $n$, which is $O(d^n)$.

- The linear layers $\boldsymbol{C}$ perform a vector-matrix multiplication of cost $k^2 O(d^n)$, this is due we have a matrix-vector product of $k^2$ involving $O(d^n)$ terms per operation.

So for every recursive step, this results to the following cost of $O(dk + d^j + k^2 d^j)$, and the final output step incurs a cost of $O(d^n ko)$. So the overall cost of the function reconstruction is:

$$kod^n + \sum_{j=1}^{n-1} \left(dk + d^j + k^2 d^j\right) = kod^n + (n-1)dk + (k^2 + 1)\frac{d^n - 1}{d - 1} = O(k^2 d^n)$$

The significantly increased cost is caused linear layers between hidden inputs, as they perform on very large symbolic expressions. It is also important to note that for the $\pi$-netV1 the hidden rank can significantly larger than the dimensions or the degree. Note that this proof can also be done by using the equations for reconstructing the $\boldsymbol{\mathcal{W}}^{[n]}$ from Chrysos et al. (2021b) and the same result will be shown.

## BACKGROUND AND NOTATION

In the , we develop a more detailed notation and the core symbols of our work.

Table 3: Core symbols and notations used in this work.

| Symbol | Dimension(s) | Definition |
|---|---|---|
| $\mathcal{N}(\mu, \sigma)$ | - | Gaussian distribution of mean $\mu$ and variance $\sigma$ |
| $d$ | $\mathbb{N}_+$ | Size of a datapoint |
| $o$ | $\mathbb{N}_+$ | Output size of the network |
| $n$ | $\mathbb{N}_+$ | Degree of the network |
| $k$ | $\mathbb{R}_+$ | width of the network or rank o the decomposition |
| $[N]$ | - | the discrete set $\{1, 2, ...N\}$ |

Table 4: Symbols on a matrix $\boldsymbol{U}$.

| Symbol | Definition |
|---|---|
| $\boldsymbol{U}^{[k]}$ | The $k^{th}$ matrix in a collection of matrices $\{\boldsymbol{U}^{[1]}, ...\boldsymbol{U}^{[n]}\}$ |
| $U_{ij}$ | the $(i, j)^{th}$ element of $\boldsymbol{U}$ |
| $\boldsymbol{u}_i^{(k)}$ | the $i^{th}$ row of $\boldsymbol{U}^{(k)}$ |
| $u_{ij}$ | the $j^{th}$ row of $\boldsymbol{u}$ equivalent to $U_{ij}$ |

Table 5: Operators.

| Symbol | Definition |
|--------|------------|
| $*$ | Hadamard product (element-wise) between two vectors or matrices |
| $\langle , \rangle$ | Inner product between two vectors |

## A.3 DEFAULT NOTATION

In an attempt to encourage standardized notation, we have included the notation file from the textbook, *Deep Learning* **?** available at `https://github.com/goodfeli/dlbook_notation/`. Use of this style is not required and can be disabled by commenting out `math_commands.tex`.

**Numbers and Arrays**

| | |
|---|---|
| $a$ | A scalar (integer or real) |
| $\boldsymbol{a}$ | A vector |
| $\boldsymbol{A}$ | A matrix |
| $\mathcal{A}$ | A tensor |
| $\boldsymbol{I}_n$ | Identity matrix with $n$ rows and $n$ columns |
| $\boldsymbol{I}$ | Identity matrix with dimensionality implied by context |
| $\boldsymbol{e}^{(i)}$ | Standard basis vector $[0, \ldots, 0, 1, 0, \ldots, 0]$ with a 1 at position $i$ |
| $\text{diag}(\boldsymbol{a})$ | A square, diagonal matrix with diagonal entries given by $\boldsymbol{a}$ |
| $\mathrm{a}$ | A scalar random variable |
| $\mathbf{a}$ | A vector-valued random variable |
| $\mathbf{A}$ | A matrix-valued random variable |

**Sets and Graphs**

| | |
|---|---|
| $\mathbb{A}$ | A set |
| $\mathbb{R}$ | The set of real numbers |
| $\{0, 1\}$ | The set containing 0 and 1 |
| $\{0, 1, \ldots, n\}$ | The set of all integers between $0$ and $n$ |
| $[a, b]$ | The real interval including $a$ and $b$ |
| $(a, b]$ | The real interval excluding $a$ but including $b$ |
| $\mathbb{A} \backslash \mathbb{B}$ | Set subtraction, i.e., the set containing the elements of $\mathbb{A}$ that are not in $\mathbb{B}$ |
| $\mathcal{G}$ | A graph |
| $Pa_{\mathcal{G}}(\mathrm{x}_i)$ | The parents of $\mathrm{x}_i$ in $\mathcal{G}$ |

**Indexing**

| | |
|---|---|
| $a_i$ | Element $i$ of vector $\boldsymbol{a}$, with indexing starting at 1 |
| $a_{-i}$ | All elements of vector $\boldsymbol{a}$ except for element $i$ |
| $A_{ij}$ | Element $i, j$ of matrix $\boldsymbol{A}$ |
| $\boldsymbol{A}_{i,:}$ | Row $i$ of matrix $\boldsymbol{A}$ |
| $\boldsymbol{A}_{:,i}$ | Column $i$ of matrix $\boldsymbol{A}$ |
| $\mathsf{A}_{i,j,k}$ | Element $(i, j, k)$ of a 3-D tensor $\mathsf{A}$ |
| $\mathsf{A}_{:,:,i}$ | 2-D slice of a 3-D tensor |
| $\mathrm{a}_i$ | Element $i$ of the random vector $\mathbf{a}$ |

### Calculus

| | |
|---|---|
| $\dfrac{dy}{dx}$ | Derivative of $y$ with respect to $x$ |
| $\dfrac{\partial y}{\partial x}$ | Partial derivative of $y$ with respect to $x$ |
| $\nabla_{\boldsymbol{x}} y$ | Gradient of $y$ with respect to $\boldsymbol{x}$ |
| $\nabla_{\boldsymbol{X}} y$ | Matrix derivatives of $y$ with respect to $\boldsymbol{X}$ |
| $\nabla_{\mathsf{X}} y$ | Tensor containing derivatives of $y$ with respect to $\mathsf{X}$ |
| $\dfrac{\partial f}{\partial \boldsymbol{x}}$ | Jacobian matrix $\boldsymbol{J} \in \mathbb{R}^{m \times n}$ of $f : \mathbb{R}^n \to \mathbb{R}^m$ |
| $\nabla_{\boldsymbol{x}}^2 f(\boldsymbol{x})$ or $\boldsymbol{H}(f)(\boldsymbol{x})$ | The Hessian matrix of $f$ at input point $\boldsymbol{x}$ |
| $\displaystyle\int f(\boldsymbol{x}) d\boldsymbol{x}$ | Definite integral over the entire domain of $\boldsymbol{x}$ |
| $\displaystyle\int_{\mathbb{S}} f(\boldsymbol{x}) d\boldsymbol{x}$ | Definite integral with respect to $\boldsymbol{x}$ over the set $\mathbb{S}$ |

### Probability and Information Theory

| | |
|---|---|
| $P(\mathrm{a})$ | A probability distribution over a discrete variable |
| $p(\mathrm{a})$ | A probability distribution over a continuous variable, or over a variable whose type has not been specified |
| $\mathrm{a} \sim P$ | Random variable a has distribution $P$ |
| $\mathbb{E}_{\mathrm{x} \sim P}[f(x)]$ or $\mathbb{E} f(x)$ | Expectation of $f(x)$ with respect to $P(\mathrm{x})$ |
| $\mathrm{Var}(f(x))$ | Variance of $f(x)$ under $P(\mathrm{x})$ |
| $\mathrm{Cov}(f(x), g(x))$ | Covariance of $f(x)$ and $g(x)$ under $P(\mathrm{x})$ |
| $H(\mathrm{x})$ | Shannon entropy of the random variable x |
| $D_{\mathrm{KL}}(P\|Q)$ | Kullback-Leibler divergence of P and Q |
| $\mathcal{N}(\boldsymbol{x}; \boldsymbol{\mu}, \boldsymbol{\Sigma})$ | Gaussian distribution over $\boldsymbol{x}$ with mean $\boldsymbol{\mu}$ and covariance $\boldsymbol{\Sigma}$ |

### Functions

| | |
|---|---|
| $f : \mathbb{A} \to \mathbb{B}$ | The function $f$ with domain $\mathbb{A}$ and range $\mathbb{B}$ |
| $f \circ g$ | Composition of the functions $f$ and $g$ |
| $f(\boldsymbol{x}; \boldsymbol{\theta})$ | A function of $\boldsymbol{x}$ parametrized by $\boldsymbol{\theta}$. (Sometimes we write $f(\boldsymbol{x})$ and omit the argument $\boldsymbol{\theta}$ to lighten notation) |
| $\log x$ | Natural logarithm of $x$ |
| $\sigma(x)$ | Logistic sigmoid, $\dfrac{1}{1 + \exp(-x)}$ |
| $\zeta(x)$ | Softplus, $\log(1 + \exp(x))$ |
| $||\boldsymbol{x}||_p$ | $L^p$ norm of $\boldsymbol{x}$ |
| $||\boldsymbol{x}||$ | $L^2$ norm of $\boldsymbol{x}$ |
| $x^+$ | Positive part of $x$, i.e., $\max(0, x)$ |
| $\mathbf{1}_{\text{condition}}$ | is 1 if the condition is true, 0 otherwise |

