# OpenReview forum: "TUCKER DECOMPOSITION FOR INTERPRETABLE NEURAL ORDINARY DIFFERENTIAL EQUATIONS"
_ICLR.cc/2024/Workshop/AI4DiffEqtnsInSci — AI4DiffEqtnsInSci @ ICLR 2024 Poster_

### Official Review · Reviewer_vG27 · 2024-02-15
**Results from numerical experiments are not very convincing**

**Rating:** 4
**Confidence:** 4

**Review:**

The authors introduced a novel framework that takes advantage of Tucker decomposition for polynomial networks to learn solutions of ODEs.

Major comments:
- Important references in the literature are currently missing, for example -- Kossaifi, Jean, Nikola Kovachki, Kamyar Azizzadenesheli, and Anima Anandkumar. "Multi-Grid Tensorized Fourier Neural Operator for High-Resolution PDEs." arXiv preprint arXiv:2310.00120 (2023). There, the authors already demonstrated the usage of Tucker decomposition for representation of network weights in neural operators. For the current manuscript, I feel like the usage of Tucker is not well motivated and the experiments are very low-dimensional and therefore not very convincing.
- It is very hard to see the uplift of TuckerNet in the ODE learning case study. Is RMSE really an effective measure? There are always some events that are better captured by other two networks and some other ones are better captured by TucketNet. I suggest authors either provide more metrics, or doing more analysis on these signals -- for example, does it make sense to look at the phase matching? See whether certain events are shifted away? There are 2 main factors in these signals --- amplitude and "peak time". I am afraid RMSE cannot speak for both, especially when the RMSE values for these three networks are very close.
- There are some strong claims in the manuscript and I do not think they are substantiated by experiments or references, e.g. "Tucker formulated polynomials could be less likely to get stuck on worst local minima."

Minor comments:
- What are "parameter-constrained" models?
- Several typos, e.g. "the we", "CDP"

---

### Official Review · Reviewer_jmmK · 2024-02-26
**TUCKER DECOMPOSITION FOR INTERPRETABLE NEURAL ORDINARY DIFFERENTIAL EQUATIONS**

**Rating:** 6
**Confidence:** 4

**Review:**

This paper talks about Tucker decomposition for ODE. The paper covers the theory of Tucker decomposition. The paper covers the complexity of the algorithm, however is unable to discuss the memory pressure. The paper also lacks the statistical comparison of the results and variability on multiple ODEs. From the workshop perspective, the paper appears to bring in a new idea, however more tests need to be done for representability.

---

### Meta-Review · Area_Chair_Uhpt · 2024-03-01

**Recommendation:** Accept (Poster)

**Metareview:**

Thanks to reviewers for their feedback. Reviewer vG27 provided good comments for authors to incorporate in their final version. I appreciate the effort authors made in this study of the advantage of Tucker decomposition in neural ODE modeling and both reviewers addressed the novelty. I vote accept (poster) for this paper.

---

### Decision · Program_Chairs · 2024-03-02

Accept (Poster)